# A Novel Cre/*lox*-Based Genetic Tool for Repeated, Targeted and Markerless Gene Integration in *Yarrowia lipolytica*

**DOI:** 10.3390/ijms221910739

**Published:** 2021-10-04

**Authors:** Qinghua Zhou, Liangcheng Jiao, Wenjuan Li, Zhiming Hu, Yunchong Li, Houjin Zhang, Min Yang, Li Xu, Yunjun Yan

**Affiliations:** Key Laboratory of Molecular Biophysics, The Ministry of Education, College of Life Science and Technology, Huazhong University of Science and Technology, Wuhan 430074, China; qinghuazhou1@126.com (Q.Z.); jiaoliangcheng@gmail.com (L.J.); wenjuanli1@163.com (W.L.); zhiming711@163.com (Z.H.); liyunchong77@hust.edu.cn (Y.L.); hjzhang@hust.edu.cn (H.Z.); ymyangmin@hust.edu.cn (M.Y.)

**Keywords:** genetic tool, Cre/*lox*, repeatedly targeted and markerless integration, *Yarrowia lipolytica*

## Abstract

The unconventional yeast *Yarrowia lipolytica* is extensively applied in bioproduction fields owing to its excellent metabolite and protein production ability. Nonetheless, utilization of this promising host is still restricted by the limited availability of precise and effective gene integration tools. In this study, a novel and efficient genetic tool was developed for targeted, repeated, and markerless gene integration based on Cre/*lox* site-specific recombination system. The developed tool required only a single selection marker and could completely excise the unnecessary sequences. A total of three plasmids were created and seven rounds of marker-free gene integration were examined in *Y. lipolytica*. All the integration efficiencies remained above 90%, and analysis of the protein production and growth characteristics of the engineered strains confirmed that genome modification via the novel genetic tool was feasible. Further work also confirmed that the genetic tool was effective for the integration of other genes, loci, and strains. Thus, this study significantly promotes the application of the Cre/*lox* system and presents a powerful tool for genome engineering in *Y. lipolytica*.

## 1. Introduction

With the development of synthetic biology, various molecular genetic tools are being increasingly designed for precise genome modification in yeasts. As a promising host, the generally regarded as safe (GRAS) yeast, *Yarrowia lipolytica*, has been employed in a wide range of fields such as protein production, metabolic engineering, degradation of *n*-alkanes, and dimorphism [1,2,3,4,5,6]. Nevertheless, only a small number of effective genetic tools have been developed for repeatedly targeted gene integration in this host, which still restricts the further exploitation and application of *Y. lipolytica*.

In traditional methods, heterologous and endogenous genes are transformed into the *Y. lipolytica* genome via the insertion of the whole plasmid at target loci or the integration of gene fragments and selection markers at zeta loci, and many copies of the zeta sequence present in the Ylt1-carrying host provide a large number of potential sites for the construction of multi-copy transformants [7,8,9]. However, the introduction of dispensable DNA fragments, especially antibiotics resistance genes, may harm the GRAS status of *Y. lipolytica* and hinder its application in the food and medicine industries. Furthermore, the bioproduction of *Y. lipolytica* usually requires multi-round integrations or co-expression of various genes [3,10,11,12,13], which are difficult to realize via these conventional methods owing to the limited availability of selection markers. Consequently, there is an urgent need to develop efficient genetic tool(s) for targeted, repeatable, and markerless gene integration in *Y. lipolytica*.

To date, several site-specific nuclease technologies have been established for genome modification in various species [14,15,16,17]. Among them, the CRISPR-Cas9 system can accomplish relatively high-efficiency gene disruption and marker-free gene integration in *Y. lipolytica* [18,19,20]. A previous study examined 17 loci for targeted and markerless gene integration in *Y. lipolytica*, among which only five loci exhibited gene integration, with the highest integration efficiency being 68.9% ± 25.5% [21]. Another study accomplished gene integration into the *PEX*10 locus in *Y**. lipolytica* via a dual CRISPR-Cas9 cleavage strategy directed by paired sgRNAs; the integration efficiencies of the homologous recombination (HR)-based method, using a circular plasmid as the repair template, and of the homology-mediated end-joining based method, using a linear template, were 16.7% ± 3.6% and 37.5% ± 8.8%, respectively [22]. These studies indicate the need for further improvements in terms of achieving more desirable gene integration efficiency (similar to gene disruption efficiencies of about 90%) via CRISPR-Cas9-based tools. Furthermore, gene insertion via homologous recombination, initiated by repair or double-stranded DNA breakages introduced by CRISPR-Cas9, usually demands the synchronous introduction of plasmid-harboring CRISPR-Cas9-related elements and a repair template [21,22], resulting in a reduction in transformation efficiency and the need for an additional procedure to remove CRISPR-Cas9-related elements.

The Cre/*lox*-site-specific recombination system derived from bacteriophage P1 is known to be appropriate for genome modification [23,24,25,26], and its mechanism of action has been clearly described [27,28,29]. Cre recombinase can recognize and bind to the *lox*P site, and can catalyze recombination, resulting in insertion, translocation, inversion, and deletion [27]. However, these reactions can retain the redundant *lox*P site in the genome, resulting in genome instability, and also making strain construction more difficult. To address this issue, some mutated *lox* sites have been successfully designed [28,29]. For example, the recombination between a left element-mutant *lox* site (LE), *lox*71, and a right element-mutant *lox* site (RE), *lox*66, can generate a double element-mutant *lox* site (LE + RE), *lox*72, and a *lox*P site [28,29]. Meanwhile, the *lox*P site can be deleted and the *lox*72 site can be retained in the genome through rational design. In particular, the *lox*72 site, which can weakly bind to Cre recombinase, is less likely to participate in subsequent recombination [28,29]. As a result, mutated *lox* sequences can exhibit enhanced controllability of recombination and accelerate the utilization of the Cre/*lox* system in yeasts [30,31,32,33]. Although this method can employ marker-free genome modification, it often presents low efficiency or is time-consuming. In addition, a few studies have attempted markerless gene integration via the Cre/*lox* system in *Y. lipolytica*. Among them, an inspired method combining the Cre/*lox* system and 26S rDNA achieved 79.2% integration efficiency in *Y. lipolytica* [34]; however, the integration loci were random and the gene copy numbers were uncontrollable, requiring a lot of time for the selection and evaluation of transformants. Moreover, to rescue the selection marker in the genome, an additional plasmid-harboring *cre* gene must be introduced into the recombinants via another transformation, and this plasmid should subsequently be recovered. These procedures require two different selection markers and further extend the experimental period.

Therefore, in the present study, a novel and effective genetic tool was designed based on the Cre/*lox* system. Only a single selection marker was adequate to enable repeated, targeted, and markerless gene integration, and the unnecessary fragments could be excised. To demonstrate the efficacy of the developed tool, the *Rhizomucor miehei* lipase gene (*rml*) was chosen as a reporter gene, and three new plasmids were constructed. A total of seven rounds of marker-recyclable integration were implemented, and the positive colony proportions reached 90.63%–100%. The experimental conditions, feasibility, and controllability of the genetic tool were evaluated and optimized. Moreover, to confirm the universality of the developed tool, another six genes (the *ire1*, *kar2*, *pdi*, *sls1*, *hac1,* and *vgb* genes) were further integrated into the *LEU2* locus, and a derived tool was successfully employed to integrate another gene (*Rhizopus oryzae* lipase gene, *rol*) into the *axp1-2* locus. The results obtained reveal that the Cre/*lox*-based genetic tool could resolve the limitations of the existing methods and bring significant advancement in the targeted modification of the genome in *Y. lipolytica*.

## 2. Results

### 2.1. Genetic Tool Design

The novel Cre/*lox*-based genetic tool comprised three plasmids, namely Cre-Y1, Cre-Y2, and Cre-Y3 (Figure 1A). Plasmid Cre-Y1 was applied to introduce a single *lox*71 fragment into the host genome DNA, and plasmids Cre-Y2 and Cre-Y3 were employed to iteratively introduce the *rml* gene into the host strain. Plasmid Cre-Y1 consisted of the *URA3* (UniProtKB, Q12724) auxotroph selection marker gene, the *cre* gene expression cassette pPOX2-cre-lip2t (pPOX2 promoter, *cre* gene, and lip2 terminator; GenBank: AJ001300.1, AB449974.1, and AJ012632.1), the ampicillin resistance gene and the origin of replication in *Escherichia coli* (Amp-ori), a 964 bp fragment located at the upstream region of the *LEU2* gene (GenBank: AF260230.1, named Upleu), and two *lox*71 sites oriented in the same direction. Plasmid Cre-Y2 was derived from Cre-Y1 by replacing Upleu and two *lox*71 sites with the *LEU2* gene homologous fragment, a *lox*66 site and its reverse complement (rc*lox*66 site), and an *rml* gene expression cassette. Plasmid Cre-Y3 was similar to Cre-Y2, but with the *lox*71 and rc*lox*71 (the reverse complement of the *lox*71 site) sites.

### 2.2. First-Round Integration to Construct the FY1 Strain

To construct strain harboring of a single *lox*71 fragment, the plasmid Cre-Y1 was inserted into the auxotrophic host Po1f (Ura^−^ Leu^−^) (Figure 1C). The transformed cell pellet was screened twice in 5 mL of MD-Leu medium (MD medium supplemented with leucine) to enrich the Ura^+^ recombinant strain Po1f/Cre-Y1 (Ura^+^ Leu^−^) and avoid interference from untransformed Po1f cells (Appendix A). Subsequently, the culture solution containing the Po1f/Cre-Y1 strain was transferred into MO-Leu-Ura medium (MO medium supplemented with leucine and uracil) for 12 h with oleic acid as the sole carbon source, which could activate pPOX2 to induce the expression of the *cre* gene. In this case, the produced Cre recombinase could mediate the deletion reactions between the two repeat *lox*71 sites in the Po1f/Cre-Y1 genome, causing excision of the Amp-ori, *URA3*, and *cre* expression cassette, and thus, further generating Ura^−^ recombinants. Meanwhile, the extra added uracil allowed Ura^−^ recombinants to grow normally. The induced cell suspension was streaked on a YPD plate to isolate a single colony with the Ura^−^ genotype, and the phenotypes of the isolated colonies were determined on YPD and MD-Leu media. As shown in Figure 2A, a total of 32 recombinants were randomly chosen and all of them could not grow on the MD-Leu plate, proving that the selection marker *URA3* was recycled successfully. Moreover, 10 positive colonies were picked up randomly and identified by genome PCR (Figure 2B), and the PCR product obtained via the primers Upleu-F3/Upleu-R3 was further sequenced. The results shown in Figure 2C indicate that only a single *lox*71 site was retained in the Upleu locus of the eventually obtained strain, named FY1 (Ura^−^ Leu^−^). This finding also reveals that all recombination and integration events occurred as expected (Figure 1C), and the adopted plasmid elements of the novel genetic tool were sufficiently suitable for *Y. lipolytica*.

### 2.3. Second-Round Integration and Analysis

To evaluate whether the designed plasmids could realize markerless gene integration, plasmid Cre-Y2 was linearized within *LEU**2* and introduced into FY1, to obtain the strain FY1/Cre-Y2 (Ura^+^ Leu^+^). As shown in Figure 1D, three *lox* sites (*lox*71, *lox*66, and rc*lox*66) were noted in the FY1/Cre-Y2 genome, and three kinds of recombination reactions could occur when Cre recombinase was expressed. In the first-type recombination (d1), deletion reaction occurred between the *lox*71 and *lox*66 sites in the FY1/Cre-Y2 genome, and the strain FY2-*rml* (Ura^−^ Leu^+^), carrying the *rml* expression cassette, was obtained. In the second-type recombination (d2), the *lox*71 and rc*lox*66 sites were combined with Cre proteins, producing a *lox*72 site and a reverse complement of the *lox*P site (rc*lox*P). The *lox*72 site was hardly bound to Cre recombinase, and the rc*lox*P site could not stably coexist with the unreacted rc*lox*66 site in the genome, which could recombine with Cre recombinase again, generating FY2-rc*rml* (Ura^−^ Leu^+^), a derived strain carrying the reverse complement of the *rml* gene expression cassette (rc*rml*). In the third-type recombination (d3), the *lox*66 and rc*lox*66 sites were recombined for an inversion reaction, forming a strain that also contained three *lox* sites that were similar to those of the FY1/Cre-Y2 strain. The recombination reaction among the three *lox* sites continued until the first or second type of recombination occurred, and the strain FY2-*rml* or FY2-rc*rml* was finally generated. Consequently, two kinds of stable strains were obtained from three types of recombination reactions in second-round integration. Here, the unnecessary and redundant fragments, including Upleu, *leu2-270*, Amp-ori, *URA3*, and the *cre* expression cassette, were completely deleted, and only the gene expression cassette and the rc*lox*66 site remained in the genome.

### 2.4. Induction Time Optimization in Second-Round Integration

In this study, the oleic acid-induced pPOX2 promoter controlled the production of Cre recombinase in MO liquid medium. Therein, the induction time affected the Cre protein concentration in the cell, and further influenced the recombination efficiency in the engineered strains. In the first-round gene integration, there were two *lox* sites in the yeast genome. The recombinants were induced for sufficiently long time periods to obtain a high proportion of positive colonies. However, in subsequent gene integrations, there were three or more *lox* sites in the genome and the recombination reaction was complex and diverse. As reported in previous studies [27,35], a high concentration of Cre protein is more likely to mediate recombination in the *lox*72 site, resulting in the removal of the *rml* gene expression cassette from the engineering strain genome. Therefore, the induction of *cre* gene expression should be controlled in an appropriate time range. In the present study, to determine the optimal induction time, the recombinant strain FY1/Cre-Y2 was incubated in MO-Ura liquid medium (MO medium supplemented with uracil) for 4–16 h and streaked onto YPD medium every 2 h (approximately one cell generation cycle). Furthermore, 32 recombinants were randomly selected from each YPD medium and then transferred onto MD and BMSY media. The proportions of positive colonies under different induction times were examined, and the results are shown in Table 1 and Figure 3A. All the selected recombinants could complete the recombination reaction process after 12 h of induction. For more accurate confirmation, 64 colonies induced for 12 h were chosen and none of them could grow in a normal manner on MD medium. Consequently, the induction time was set as 12 h, which could be further increased if the proportion of positive colonies decreased in multi-round integrations.

A total of 10 positive colonies were randomly chosen for identification via PCR sequencing (Figure 3B). The *rml*/rc*rml* expression cassette and rc*lox*66 site in all of the selected colonies matched with the genome of the FY2-*rml*/FY2-rc*rml* strains (Figure 3C), demonstrating that the second-round gene integration was performed according to the predictive analysis (Figure 1D). In addition, for the convenience of subsequent protein production evaluation, strains that did not carry the rc*rml* expression cassette were employed for further gene integration, because the existing plasmids and method [36] are more applicable to construct strains harboring only *rml* expression cassettes.

### 2.5. Iterative Insertion of Cre-Y3 and Cre-Y2 Plasmids

To achieve third-round integration, plasmid Cre-Y3 was utilized to transform the FY2-*rml* strains, and the schematic diagram is presented in Figure 1E. Similar to the second-round integration, there were three mutated *lox* sequences (rc*lox*66, rc*lox*71, and *lox*71) and a *lox*72 site in the FY2-*rml*/Cre-Y3 (Ura^+^ Leu^+^) genome, resulting in three different recombination reactions and eventually generating two genotypes of the recombinant strains FY3-2*rml* (Ura^−^ Leu^+^) and FY3-*rml*-rc*rml* (Ura^−^ Leu^+^). Subsequently, FY3-2*rml* and FY3-*rml*-rc*rml* were distinguished by PCR amplification with the primer pairs KNC-F2/Apa-R6 and SXB-R2/Apa-R3 (Appendix A). The obtained products were further sequenced and the results (Appendix A) were consistent with the prediction presented in Figure 1E. Thus, another copy of the *rml* gene was targeted on the *LEU2* locus and the unnecessary DNA fragments were excised in the third-round integration.

Similarly, plasmid Cre-Y2 was introduced into the FY3-2*rml* to generate the FY4-3*rml*/FY4-2*rml-*rc*rml* (Ura^−^ Leu^+^) strains, and plasmid Cre-Y3 was used to transform FY4-3*rml* to construct the FY5-4*rml*/FY5-3*rml-*rc*rml* (Ura^−^ Leu^+^) strains (Appendix A). Subsequently, Cre-Y2 and Cre-Y3 were iteratively integrated into the corresponding recombinant strains to obtain FY6-5*rml*/FY6-4*rml-*rc*rml* (Ura^−^ Leu^+^) and FY7-6*rml*/FY7-5*rml-*rc*rml* (Ura^−^ Leu^+^). The genome PCR identification results (Appendix A) prove that repeated, targeted, and markerless gene integration was successfully implemented by the novel genetic tool.

### 2.6. Increasing Induction Time in Sixth and Seventh Rounds of Integration

As shown in Table 1, the positive colony proportions slightly decreased in multi-round integration processes, which might have been due to the following possibility: an additional *lox*72 site or rc*lox*72 site (reverse complement of the *lox*72 site) could have been inserted into the genome in each round of gene integration, except in the first-round integration, and all of the *lox* sites were able to bind to Cre recombinase [27,28,29]. Thus, the *lox*72 site might have disturbed the recognition between the LE or RE sites and Cre recombinase, resulting in less recombination frequency during the induction process and a decrease in efficiency in multi-round integrations. Nevertheless, an increase in induction time caused higher accumulation of Cre protein in the cell, and more Cre recombinase could recognize LE or RE sites and mediate effective recombination.

Accordingly, the transformants were induced for longer time periods to obtain higher recombination efficiency in the sixth-round integration. As a result, the proportion of positive colonies was enhanced, and 16 h was adequate to complete recombination in the engineered strains (Table 1 and Appendix A). In the seventh-round integration, 16 h was employed as the induction time and all of the chosen colonies were unable to grow on MD medium (Appendix A). Therefore, recombination efficiency might be roughly inversely proportional to the integration round; however, this problem could be alleviated by increasing the induction time in MO-Ura liquid medium. Finally, seven rounds of gene integration were implemented in this study, and the positive colony proportions reached 90.63%–100% with an induction time of 12–16 h (Table 1). In fact, the realized integration rounds were adequate to meet the demands of the construction of engineering strains, and the potential of the developed genetic tool could also be explored in future research.

### 2.7. Protein Expression Level and Growth Characteristics Evaluation

Recombinants harboring different *rml* gene copies were cultured in BMSY liquid medium for protein production, and the results are presented in Figure 4A. The recombinant strains FY5-4*rml* 31# showed the highest RML expression level, while the lipase activities of FY6-5*rml*/FY6-4*rml*-rc*rml* and FY7-6*rml*/FY7-5*rml*-rc*rml* strains were slightly lower. Thus, gene integration was terminated in the seventh round. Moreover, several colonies were chosen as representatives for each type of the recombinant strain, and their *rml* gene copies were determined via qPCR. As shown in Table 1, all of the transformants carried the same gene copy number as previously predicted, indicating that no *rml* expression cassette was excised from the genome, and also confirming the excellent controllability and stability of the genetic tool.

To evaluate the protein expression level of the recombinant strains generated via the developed genetic tool, the plasmids hp12d-5*rml* and hp12d-6*rml* were constructed according to the previous method [36]. Subsequently, the hp12d-n*rml* plasmids (n = 1–6) were linearized and introduced into Po1f to form the recombinant strains Po1f/hp12d-n*rml* (carrying n copies of the *rml* expression cassette, n = 1–6). Then, Po1f/hp12d-n*rml* (n = 1–6) and engineered strains from seven rounds of gene integration were, respectively, inoculated into the BMSY medium under the same culture conditions, and their lipase activity was measured (Figure 4B), which reveals that the two types of recombinant strains showed similar protein expression levels. Therefore, it can be concluded that genome modification via the developed genetic tool did not affect the target protein production of the engineering strains. When compared with the previous method [36], the developed genetic tool has many advantages such as the utilization of reusable markers, the requirement of three plasmids, the absence of gene redundancy, and the capacity for flexible direction of the gene expression cassette. Moreover, the novel genetic tool could assemble engineering strains harboring any number of copies of the *rml* gene, only requiring increases/decreases in integration rounds.

Furthermore, the strains Po1f, FY1, FY5-4*rml* 31#, and FY7-6*rml* 13# were cultivated in YPD and BMSY media, respectively, and their growth characteristics were tested by measuring the optical density at 600 nm (OD_600_) at different incubation times (Figure 4C,D). The results reveal no observable differences in the growth curves of the four types of strains, indicating that the genome modification in the *LEU**2* locus via the Cre/*lox*-based genetic tool did not influence cell growth.

### 2.8. Comparison of Two-Genotype Recombinant Strains

Two genotypes of the recombinant strains were generated in each round of integration (except for the first round), and their numbers are presented in Table 1. The rates of the two forms of strains were approximately equal, suggesting that there was nearly equal possibility to choose a strain with a certain genotype. The colonies with high lipase activities were selected for protein production evaluation. The results shown in Table 1 also imply that the protein expression levels of the recombinant strains in each round of integration were similar, which is confirmed in Figure 4A. In contrast, some previous studies have reported that the relative arrangement of multiple genes might affect the corresponding transcription and protein production in host cells [37,38]. Thus, a total of 30 positive colonies from the third-round integration were identified via the primer pairs KNC-F2/Apa-R6 and SXB-R2/Apa-R3 (data not shown), and their lipase activities were measured. The results presented in Figure 4E show that the RML activity of each recombinant randomly changed within a proper range. Moreover, transcriptional level analysis further indicates that there was no significant difference in the transcriptional quantity of *rml* mRNA isolated from FY3-2*rml* and FY3-*rml*-rc*rml* cells (Figure 4F). Thus, the relative direction of *rml* expression cassettes had no observable effect on protein production in this study, which might have been due to the difference in the host type. Accordingly, both the genotypes of the strains were concluded to be suitable for RML production in *Y. lipolytica*.

### 2.9. Successfully Expanding to Six Different Genes and axp1-2 Locus

To evaluate the universality of the novel genetic tool, six genes with varied lengths and two resources were additionally integrated into the engineered strain FY5-4*rml*. The genes *ire1*, *kar2*, *pdi*, *sls1*, and *hac1* are the homologous genes of *Y. lipolytica*, and *vgb* is the coding gene of *Vitreoscilla* hemoglobin. These genes may differentially affect RML expression levels, which determine the number of integration rounds required to increase their gene dosage to facilitate the expression of RML. The corresponding plasmids were derived from Cre-Y2 and Cre-Y3 by replacing the *rml* gene with different genes (Appendix A), and then used to transform FY5-4*rml*. The subsequent integration processes were similar to those of Cre-Y2 and Cre-Y3, and the proportions of positive colonies were all above 90% (Table 2), confirming the availability of the developed genetic tool for repeated, targeted, and markerless integration of different genes.

Furthermore, a derived tool was established to verify whether the developed genetic tool could be applied for other genes, integration loci, and strains. For this purpose, three plasmids containing *rol* gene were, respectively, constructed based on Cre-Y1, Cre-Y2, and Cre-Y3 (Appendix A), and integrated into a defective acid extracellular protease gene (*AXP*, GenBank: XM_500538.1) locus (*axp1-2*) of *Y. lipolytica* Po1h (Ura^−^ Leu^+^). The transformation, screening and induction, isolation of positive colonies, and identification processes were similar to those employed for *rml* gene integration into the *leu2**-270* locus of the Po1f strain. First, the Cre-axp1 plasmid was linearized by *Nsi*I and inserted into the Upaxp locus (a 1059 bp fragment located at the upstream region of *AXP gene*) to generate a strain carrying a single *lox*71 site. Then, the Cre-axp2 and Cre-axp3 plasmids were digested with *Cla*I and iteratively introduced into the *axp1-2* locus for repeated, targeted, and markerless integration of *rol* in Po1h. As a result, three rounds of integration were conducted and all of the integration rates were above 96%. These findings reveal that such extension of the novel genetic tool is feasible and convenient.

## 3. Discussion

In the present study, a Cre/*lox*-based genetic tool was precisely and skillfully designed. The plasmids Cre-Y2 and Cre-Y3 adopted two inversely oriented *lox* sites, and the *URA3* marker gene could not be removed from the genome if the strain did not undergo integration process as expected. Thus, the Ura^+^ strain could be distinguished by isolating positive colonies on the MD medium. It must be noted that except for the *cre* gene and the *lox* sites, all of the genetic tool elements were nonspecific and replaceable. Any DNA sequence could be chosen as the homologous fragment as long as no gene was disrupted, and the length of the homologous fragment was very flexible because of the excision of unnecessary sequences. The *rml* gene in plasmids Cre-Y2 and Cre-Y3 could be replaced by heterologous and endogenous genes of *Y. lipolytica* to accomplish repeated markerless integration, and the plasmid construction process was simple and convenient. In addition, the Cre/*lox*-based genetic tool could also be applied to other genes and loci if the corresponding plasmid elements were available. 

Recently, several genetic tools have been developed for the markerless gene integration of *Y. lipolytica* based on the CRISPR-Cas9 system, Cre/*lox* system and Ura/5-fluoroorotic acid resistance method (Appendix A) [19,21,22,34]. The integration efficiency markedly fluctuates with different genome loci and single-guide RNAs (sgRNAs) for CRISPR-Cas9-mediated gene integrations [19,21]. According to most of the existing Cre/*lox*-based methods [19,34], the DNA fragment can be integrated into the host genome via gene insertion or gene replacement. However, insertion events may introduce unnecessary DNA segments into the genome, whereas replacement events, which occur occasionally, can avoid this issue, but require substantial screening of target transformants. Furthermore, while the Ura/5-fluoroorotic acid resistance method (Ura pop-in/pop-out method) can also accomplish markerless gene integration, reverse mutation is difficult to avoid in the marker recovery process. In addition, new plasmids carrying different sgRNA or homologous fragments must be constructed to repeatedly realize gene integration in a specific genome locus via the three above-mentioned genetic tools. These limitations can be completely overcome using the novel genetic tool designed in the present study. The *rml* gene is integrated into the host genome via an efficient, targeted, and repeatable gene insertion event, and the introduced dispensable sequences can be deleted via the recombination reaction of the *lox* sites. Moreover, despite the occurrence of inefficient insertion events in the integration process, a few recombinants can be enriched via double auxotroph screening. The existing methods, based on the CRISPR-Cas9 and Cre/*lox* systems, require two different selection markers and an additional plasmid recovery procedure [19,21,22,34]. In contrast, only a single marker is required for the novel Cre/*lox*-based genetic tool, and the processes of gene integration, marker reuse, and plasmid recovery can be combined and accomplished simultaneously, which can significantly reduce the experimental period and workload. Overall, the novel genetic tool developed in this study can overcome many limitations such as low HR rates, limited selection markers, random integration sites, and additional plasmid recovery.

In conclusion, a powerful and efficient Cre/*lox*-based genetic tool for repeated, targeted, and markerless gene integration was developed in this study. In practice, only a single selection marker can enable the iterative integration of various genes into the genome, which can facilitate genetic modifications of the host strains. Furthermore, in each round of integration, the unnecessary segments are removed, the relative direction of each gene expression cassette is controllable, and the gene recombination on genome is foreseeable. These characteristics can significantly decrease gene redundancy in the engineering strain, making it convenient for the use of the host strain in pharmaceutical, food, and other health-related areas. Besides, the developed genetic tool is also suitable for various other genes and loci. The findings of this study significantly broaden the application of the Cre/*lox* system and strengthen the capabilities for genome modification in *Y. lipolytica*.

## 4. Materials and Methods

### 4.1. Strains and Media

*E**. coli* TOP10 and TOP10 F’ (Shanghai Weidi Biotechnology Co., Ltd., Shanghai, China) were applied for plasmid amplification. The plasmid hp12d-*rml* [36] and auxotrophic *Y. lipolytica* Po1f [39] and Po1h [40] were, respectively, utilized for plasmid construction and transformation. The T4 DNA ligase, Phanta Max Master Mix polymerase, Rapid Taq Master Mix polymerase, and ClonExpress II One Step Cloning Kit were obtained from Vazyme Biotech Co., Ltd. (Nanjing, China) and used for gene amplification and genome identification. The *E. coli* and *Y. lipolytica* strains were, respectively, cultured at 37 °C and 28 °C. All strains and plasmids used in this study are listed in Appendix A. The LB, YPD, MD, and BMSY media have been previously described [36]. The MO medium (20 mL/L oleic acid, 0.5 mL/L Tween 80, 13.4 g/L yeast nitrogen base, with no amino acids and 0.4 mg/L biotin), was derived from MD medium, and leucine (final concentration, 262 mg/L) or uracil (final concentration, 22.4 mg/L) was added for auxotrophic strains.

### 4.2. Plasmid Construction

Appendix A illustrates plasmid Cre-Y1 construction. The pPOX2 promoter, *cre* gene, tandem fragment of lip2t and *lox*71, and fusion segment of Upleu and *lox*71 were, respectively, cloned via corresponding primer pairs (Pox2-F1/Pox2-R1, Cre-F1/Cre-R1, lip2t-F1/lip2t-R1, and Upleu-F1/Upleu-R1). The four amplified products were assembled by overlap extension PCR and then double-digested with *Nde*I/*Apa*I. Subsequently, the digested product was inserted into *Nde*I/*Apa*I-digested plasmid hp12d-*rml* to construct plasmid 1296*-cre*. Furthermore, the *URA3* gene (amplified using Ura-F1/Ura-R1) was subcloned into *Nde*I-linearized 1296*-cre* by HR, generating plasmid Cre-Y1. All the plasmids were confirmed via DNA sequencing by TsingKe Biological Technology Co., Ltd. (Wuhan, China), and all of the primers and *lox* sites used in this study are presented in Appendix A. 

As shown in Appendix A, the plasmids Cre-Y1 and hp12d-*rml* were, respectively, double-digested using *Apa*I/*Nhe*I and *Apa*I/*Xba*I, and the two digestion products with the same cohesive ends were ligated to generate the plasmid hp12d-*rml-cre*. Subsequently, the gene segment lip2t-*lox*66 was cloned by lip2t-F1/lip2t-R2 and recombined into *Avr*II/*Mlu*I-digested hp12d-*rml-cre*, forming hp12d-*rml-cre*-*lox*66. Furthermore, the fusion fragment of rc*lox*66 and partial *LEU**2* was amplified by leu-F2/leu-R2 and recombined with *Nde*I/*Apa*I-opened hp12d-*rml-cre*-*lox*66 to generate plasmid Cre-Y2. Similarly, the gene segment lip2t-rc*lox*71 was amplified by lip2t-F1/lip2t-R3 and used for the construction of hp12d-*rml-cre*-rc*lox*71, and hp12d-*rml-cre*-rc*lox*71 was recombined with *lox*71-partial *LEU**2* to obtain plasmid Cre-Y3 (Appendix A). 

The *rml* gene of the Cre-Y2 and Cre-Y3 plasmids was replaced with six different genes (*ire1*, *kar2*, *pdi*, *sls1*, *hac1*, and *vgb*) to assemble the corresponding plasmids, and their structures are presented in Appendix A. Moreover, the homologous fragments of the Upleu, *LEU**2,* and *rml* genes were, respectively, substituted with Upaxp, *axp1-2*, and *rol* to generate the plasmids Cre-axp1, Cre-axp2, and Cre-axp3 (Appendix A). The above-mentioned procedures were achieved by conventional PCR amplification, enzyme digestion, and ligation.

### 4.3. Yeast Transformation, Screening and Induction

The integration process was performed as follows (Appendix A). The *Sal*I-linearized plasmid Cre-Y1 was integrated into the Po1f genome following the previously described method [41]. Then, the transformants were screened twice in 5 mL of MD-Leu liquid medium. Subsequently, the derivatives were inoculated into MO-Leu-Ura liquid medium for inducible expression of the *cre* gene. To isolate single colonies, the cell suspension solution was streaked onto a YPD plate, and the selected colonies were, respectively, transferred onto YPD and MD-Leu plates for phenotype verification, where the positive colonies could only grow on YPD medium. In addition, the genomic DNA of positive colonies was extracted for further identification via PCR amplification with appropriate primers and DNA sequencing by TsingKe Biological Technology Co., Ltd. (Wuhan, China). Thus, the FY1 strain was obtained. The subsequent integration was almost similar to the first-round integration; however, Cre-Y2 or Cre-Y3 plasmid was linearized by *Apa*I, leucine was not required in MD or MO medium, and YPD medium was replaced with BMSY medium containing 10 mL/L glyceryl tributyrate to select colonies with high RML activity.

### 4.4. Shaking Flask Culture and Lipase Activity Determination

Transformants harboring clear and large transparent halos were inoculated into 5 mL of YPD for 24 h. Then, the cultures were transferred into 500-mL shake flasks containing 30 mL of BMSY medium and incubated for 120 h to produce RML. The RML activity in the cell culture supernatant was evaluated using a previously described method [36]. All of the experiments were performed in triplicate, and the results were analyzed by ANOVA with Duncan’s multiple range test at *p* < 0.05.

### 4.5. Determination of Gene Copy Number and Transcriptional Analysis

Based on the protocol of ChanQ^TM^ Universal SYBR qPCR Master Mix (Vazyme Biotech Co., Ltd., Nanjing, China), the *rml* gene copy numbers were determined by real-time qPCR performed using the StepOnePlus apparatus with StepOne software version 2.3 (Applied Biosystems, Foster City, CA, USA). The endogenous *act1* was set as the reference gene, and the plasmids pMD19-*act1* and hp4d-*rml* [36] were utilized for establishing standard curves, respectively. The *rml* gene transcriptional level analysis was conducted by TsingKe Biological Technology Co., Ltd. (Wuhan, China). Each sample was collected in triplicate.

## Figures and Tables

**Figure 1 ijms-22-10739-f001:**
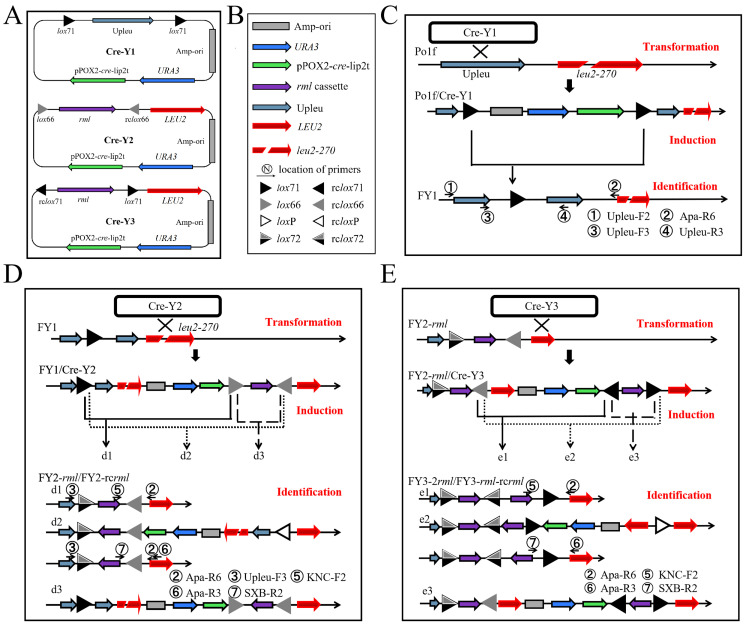
(**A**) Plasmid design of the novel genetic tool. (**B**) The legend of Figure 1. (**C**) Schematic diagram of first-round integration to obtain the FY1 strain. (**D**) Schematic diagram of second-round integration to generate the FY2-*rml*/FY2-rc*rml* strain. (**E**) Schematic diagram of third-round integration to construct the FY3-2*rml*/FY3-*rml*-rcr*ml* strain. Primers used for PCR amplification are labeled in the figure and described in Appendix A.

**Figure 2 ijms-22-10739-f002:**
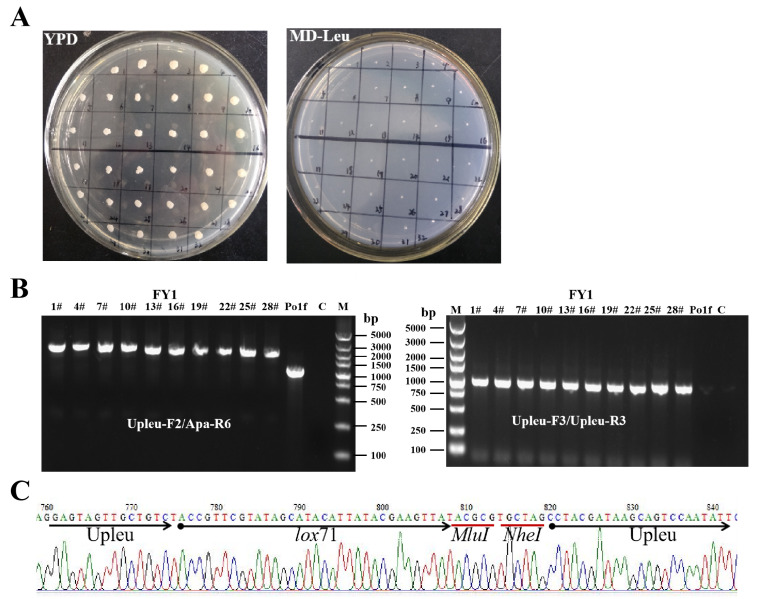
(**A**) Isolation of positive colonies in the first-round integration. A total of 32 recombinants were randomly selected and inoculated onto YPD and MD-Leu media, and none of them could grow on MD-Leu medium. (**B**) PCR amplification to identify positive colonies in the first-round integration. Lane M, Marker; lane C, ddH_2_O. PCR products of about 2400 bp (corresponding to Upleu-*lox*71-Upleu-partial *leu2-270* in FY1 genome) and about 1300 bp (corresponding to Upleu-partial *leu2-270* in Po1f genome) were, respectively, amplified by primer pair Upleu-F2/Apa-R6. The PCR product corresponding to Upleu-*lox*71-Upleu (about 1000 bp) was obtained using primers Upleu-F3/Upleu-R3 in the FY1 strain; however, no product was obtained in the Po1f strain. (**C**) Partial sequencing result for the Upleu-*lox*71-Upleu PCR product from the FY1 strain.

**Figure 3 ijms-22-10739-f003:**
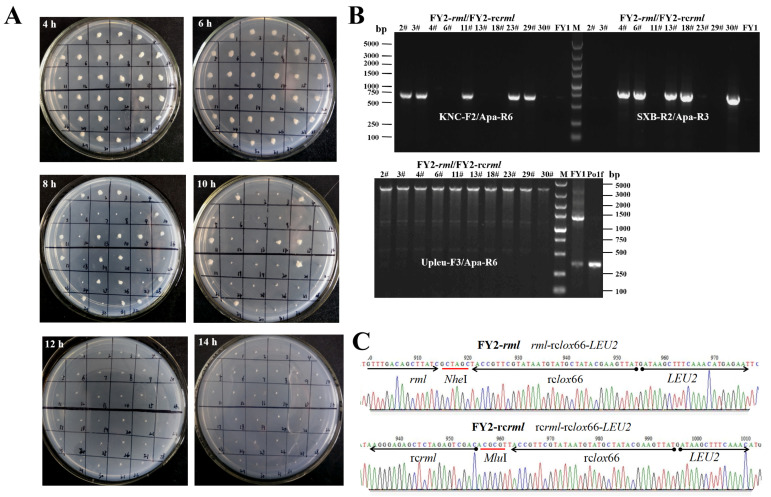
(**A**) Positive colonies induced for 4–14 h in the second-round integration. The corresponding BMSY plates are not shown. (**B**) PCR amplification to identify positive colonies in the second-round integration. Lane M, Marker. The fragment partial *rml*-rc*lox*66-partial *LEU**2* (about 650 bp) was amplified with the primer pair KNC-F2/Apa-R6 only in FY2-*rml* strains. The segment partial rc*rml*-rc*lox*66-partial *LEU**2* (about 800 bp) was cloned by SXB-R2/Apa-R3 only in FY2-rc*rml* strains. Different PCR products were obtained in four types of strains via amplification with Upleu-F3/Apa-R6: Upleu-*lox*72-*rml/*rc*rml*-rc*lox*66-partial *LEU**2* (about 3500 bp) in FY2-*rml*/FY2-rc*rml*, Upleu-*lox*71-Upleu-partial *LEU**2* (about 1300 bp) in FY1, and Upleu-partial *LEU**2* (about 300 bp) in the FY1 and Po1f strains. (**C**) Partial sequencing result of the PCR products *rml*-rc*lox*66-*LEU2* from FY2-*rml* and rc*rml*-rc*lox*66-*LEU2* from FY2-rc*rml* strains.

**Figure 4 ijms-22-10739-f004:**
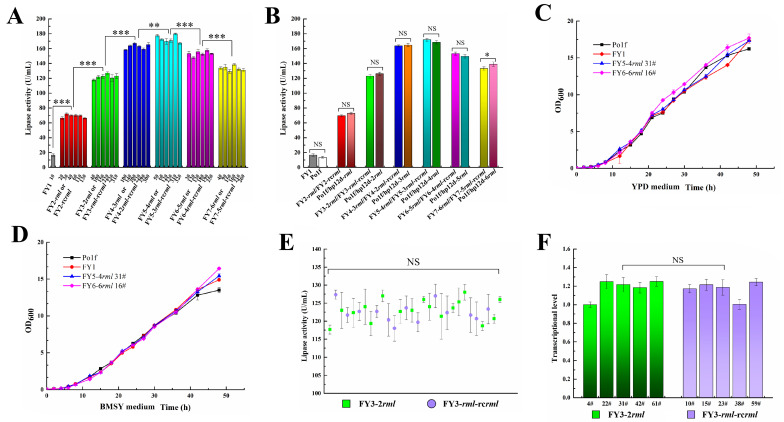
(**A**) Lipase activity of recombinant strains harboring different copies of *rml* expression cassettes. (**B**) Comparison of lipase activity of recombinant strains obtained via different methods. Data are the mean values of each kind of recombinant strain. The strains Po1f and FY1 show low lipase activity owing to expression of autologous lipase genes. Weakly significant difference is observed between the strains FY7-6*rml*/FY7-5*rml*-rc*rml* and Po1f/hp12d-6*rml* (*p =* 0.044). (**C**) Lipase activity of transformants from third-round integration. (**D**) Transcriptional level analysis of *rml* mRNA from FY3-2*rml* and FY3-*rml*-rc*rml*. The endogenous *act1* was adopted as a reference gene, and the *rml* transcriptional level of FY3-2*rml* 4# was defined as 1. (**E**) Growth curves of Po1f, FY1, FY5-4*rml* 31#, and FY7-6*rml* 13# in YPD medium. (**F**) Growth curves of Po1f, FY1, FY5-4*rml* 31#, and FY7-6*rml* 13# strains in BMSY medium. NS, *, **, and *** indicate non-significant, *p* < 0.05, *p* < 0.01, and *p* < 0.001, respectively.

**Table 1 ijms-22-10739-t001:** Positive colony proportion, genotype rate, and *rml* copy number determined in this study.

Integration Round	Induction Time (h)	Positive Colonies/All Colonies	Proportion (%)	Selected Colonies		
				Genotype	Rate	*rml* Copy Number
1	12	32/32	100	FY1	-	0
2	4	2/32	6.25	FY2-*rml*, FY2-rc*rml*	5/5	0.93
	6	2/32	6.25			
	8	14/32	43.75			
	10	24/32	75			
	12	32/32	100			
	14	32/32	100			
3	12	32/32	100	FY3-2*rml*, FY3-*rml*-rc*rml*	17/13	1.99
4	12	31/32	96.88	FY4-3*rml*, FY4-2*rml*-rc*rml*	4/2	3.10
5	12	29/32	90.63	FY5-4*rml*, FY5-3*rml*-rc*rml*	3/3	4.08
6	12	17/32	53.12	FY6-5*rml*, FY6-4*rml*-rc*rml*	4/2	5.12
	14	28/32	87.5			
	16	32/32	100			
7	16	32/32	100	FY7-6*rml*, FY7-5*rml*-rc*rml*	3/3	5.94

**Table 2 ijms-22-10739-t002:** Integration rates of various genes, loci and strains in this study.

Gene	GenBank	Lenth (bp)	Integration Round	Positive Colonies/All Colonies	Proportion (%)
Integration locus: *LEU**2* gene of FY5-4*rml*; induction time: 16 h					
*ire1*	XM_002142952.1	3663	1	32/32	100
*kar2*	U63136.1	2013	1	32/32	100
			2	32/32	100
			3	32/32	100
*pdi*	XM_503481.1	1515	1	32/32	100
			2	32/32	100
			3	31/32	96.88
			4	31/32	96.88
*sls1*	Z50154.1	1281	1	32/32	100
			2	32/32	100
*hac1*	XM_500811.1	918	1	32/32	100
*vgb*	AF292694.1	441	1	32/32	100
			2	32/32	100
Integration locus: *axp1-2* of Po1h; induction time: 12 h					
*rol*	AF229435.1	1101	1	31/32	96.88
			2	32/32	100
			3	32/32	100

## Data Availability

Data are contained within the article or Appendix A.

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
