# Peer review of "A Novel Cre/lox-Based Genetic Tool for Repeated, Targeted and Markerless Gene Integration in Yarrowia lipolytica"

_ijms, 2021, doi:10.3390/ijms221910739_

Round 1
Reviewer 1 Report
The manuscript by Zhou et al. describes a novel cre/lox-based genetic tool for Y. lipolytica. They reported a new cre/lox-based tool where they successfully demonstrated up to seven rounds of gene integration with higher than 90% efficiency per round. They also showed generalization towards other genes and other genomic loci to further strengthen their case. The work is well described and could be of interest to the research community. Still, I would like the authors to present data on generalization of their methods rather than a simple summary in Table 2. They can put these data in Supplement if space is an issue.
Other minor comments are as follows:
- Line 87: experiment conditions -> experimental conditions
- Line 101: convergently oriented lox -> lox sites oriented in the same direction
- Line 247: lane C is missing in Figure 5b
- Line 265: FY4-2rml -> FY4-3rml
- Figure 6: there is a light blue domain in 3rd panel (immediately after 'b' recombination) but not sure where that domain came from
Author Response
Response to Reviewer 1 Comments
Dear Editor and Reviewer:
We deeply appreciate the time and effort you have spent in reviewing our manuscript entitled “A Novel and Effective Cre/lox-based Genetic Tool for Repeated, Targeted and Markerless Gene Integration (ID: IJMS-1370457)”. We have carefully considered all the suggestions and comments, and made corresponding corrections via the "Track Changes" function in the revised manuscript. We do hope that the revisions will make our manuscript more acceptable for its publication. Herein, we addressed in details all the revisions according to the reviewer’ comments below:
Reviewer #1:
Point 1: The manuscript by Zhou et al. describes a novel Cre/lox-based genetic tool for Y. lipolytica. They reported a new Cre/lox-based tool where they successfully demonstrated up to seven rounds of gene integration with higher than 90% efficiency per round. They also showed generalization towards other genes and other genomic loci to further strengthen their case. The work is well described and could be of interest to the research community. Still, I would like the authors to present data on generalization of their methods rather than a simple summary in Table 2. They can put these data in Supplement if space is an issue.
Response 1: Thanks for the reviewer’s valuable comments.
In Table 2, the developed genetic tool was successfully expanded to six different genes and axp1-2 locus of Y. lipolytica Po1h. These tests were performed to confirm that our genetic tool is adequate for various genes, loci and strains, and their integration process were similar to the seven rounds of gene integration in the manuscript. Compared to the seven rounds of rml gene integration, the corresponding plasmid construction, transformation, screening and induction, isolation of positive colonies, and identification processes of Table 2 were more complicated (there are 16 rounds of gene integration in Table 2), and we were also worried about that these data would make the manuscript unfocused and chaotic. Thus, to make the manuscript more concise and readable, we only present the key information in Table 2.
Besides, to make the integration process of genes and loci in Table 2 more easy to understand and deduce, we have added some detailed description in Figure S4 of Revised Supplementary Materials.
Point 2: Line 87: experiment conditions -> experimental conditions.
Response 2: Thanks for the reviewer’s suggestion. We have made corresponding revision. Please see Line 91 in the revised manuscript.
Point 3: Line 101: convergently oriented lox -> lox sites oriented in the same direction.
Response 3: Thanks for the reviewer’s suggestion. In combination with the opinions of another reviewer, we have re-described the content of the section “2.1. Genetic tool design”, and the “convergently oriented lox” has been deleted when we rewrote this section in the revised manuscript.
Point 4: Line 247: lane C is missing in Figure 5B.
Response 4: Thanks for the reviewer’s comment. We are very sorry for the mistake that there is no “lane C” in this figure. As the alternative to “lane C, ddH2O”, “lane FY1” and “lane Po1f” were set as control groups. Meanwhile, in combination with the opinions of another reviewer, we redesigned and integrated the original figure 1, 2, 4, and 6 into a single figure (Figure 1), and updated the serial number of all figures. Please see Figure 3B in the revised manuscript.
Point 5: Line 265: FY4-2rml -> FY4-3rml.
Response 5: Thanks for the reviewer’s suggestion. We have made corresponding revision. Please see Line 283 in the revised manuscript.
Point 6: Figure 6: there is a light blue domain in 3rd panel (immediately after 'b' recombination) but not sure where that domain came from.
Response 6: Thanks for the reviewer’s comment. We are very sorry for the error. We have deleted the light blue domain. Please see Figure 1E in the revised manuscript.
In the end, we are very grateful for the reviewer and the editor for their kindness!

Reviewer 2 Report
The manuscript describes application of Cre/lox system in Yarrowia lipolytica for repeated exogenous DNA integration by recycling of the genetic marker for transformants selection. The experiments are well designed and manuscript provides enough data.
Although the authors claim: “108 - All of the tool elements were flexible and nonspecific; thus, the Cre/lox-based genetic tool was appropriate for different species.”, the tool was tested/applied only in Yarrowia lipolytica. Therefore, such and similar statements should be removed from the manuscript. Also, the title of the manuscript should contain “… in Yarrowia lipolytica”. I also suggest removing “effective” from the title – it is understood that published genetic tool should be effective.
The Section “2.1. Genetic tool design and integration process analysis” is not very informative and it does not provide information on “integration process analysis” (so this should be removed from the section title). In addition, this section (2.1) is kind of a summary of other sections, especially 2.2. and 2.3.. Therefore, the text and figures in sections 2.2 and 2.3 are partially redundant with those in section 2.1.. Actually, the figures 2, 4 and 6 are redundant with the right part of the Figure 1. Taken together, my recommendation is to merge experimental system (Figs 1, 2, 4, 6) in a single section/figure with clear and nonrepetitive explanation. (Plasmids may be presented on separate fig.)
The experimental system (Figs 1, 2, 4, 6) is interesting and well designed but it can be more precisely and more clearly presented. The figures have several issues, e.g.:
FIG1
Different terms are used in picture and in corresponding text. It is not clear what is “initial strain” (not indicated in Fig). Same for “transitional strain”. The same and convenient strain names should be used in text and Figs.
99 – “Cre-Y1 consisted of a yeast selection marker, cre gene expression cassette, bacterial element (antibiotic resistance gene and origin of replication), homologous fragment of target locus in the genome (HF-A), and two convergently oriented lox mutants. The cre expression cassette should adopt an inducible promoter to control the production of Cre recombinase. Cre-Y2 was derived from Cre-Y1 by replacing HF-A and two convergently oriented lox mutants with another homologous fragment of target locus (HF-B), a lox mutant and its reverse complement (rclox), and a target gene expression cassette. To avoid excising original segments from the genome, HF-B should be close to HF-A. Cre-Y3 was similar to Cre-Y2, but with the corresponding lox mutants of Cre-Y2 (e.g., LE for Cre-Y2 and RE for Cre-Y3).”
It should be clear that yeast selection marker is “Ura3” (are dominant and recessive genes correctly indicated? E.g. dominant URA3; recessive ura3?). Are “bacterial elements” correctly written? HF-A and HF-B are not indicated in fig etc – the text is confusing and it barely follows Fig1. I would rather use “mutated lox sequence” than “lox mutant”.
118 – 140 – text is redundant with 99-109 as well as with Figure 1 caption.
FIG2, FIG4, FIG6
The recombining sequences should be drawn in the same orientation (otherwise the picture is confusing)
The orientation of the plasmid sequence, after integration should be checked
The legend is mostly dispensable/redundant – sequences are indicated in upper pictures – Basically the same legends in those 3 Figs.
49 – “gene integration” – plasmid integration or gene replacement?
52 – “gene integration efficiency “ – plasmid integration or gene replacement (ends-in or ends-out HR)? What is efficiency of targeted plasmid integration in this manuscript?
54 - Besides, gene insertion via HR-mediated double-strand break (DSB) repair usually demands synchronous introduction of two plasmids (why? This should be further elaborated) harbouring different selection markers [21,22], resulting in reduction of transformation efficiency and additional procedure for plasmids recovery. – “gene insertion via HR…” – ends-in or ends-out mechanism?
63 – “disturb further recombination” – please explain
63 – “mutational lox sites” – or mutated?
69 – “lox mutants” – or mutated lox sequences? Etc…
83 – “integration, and all unnecessary fragments could be totally excised.” – does the lox site retain in the genome, is it necessary?
85 – “and three new plasmids were constructed.” Is this an important information in this context?
84 – “important lipase gene” – which gene?
85 – qualitative or quantitative reporter gene?
106 – it is not clear what are “original segments from the genome”
108 – flexible and nonspecific - ?
In my opinion it is appropriate to write e.g “strain Po1f” and “plasmid Cre-Y1”, not only “Po1f” or “Cre-Y1”
“target gene” and “target gene expression cassette” – in terms of genetic (homologous) recombination “target gene” is usually used for gene (or sequence) in the host genome which is targeted by integration of transforming DNA (plasmid or linear DNA fragment)
164 – “Ura3” – URA3 (if it is dominant, functional gene)
178 –“ (about 900 bp)” – there is no such band in Fig 3
…
etc
As I have already mentioned, the manuscript is interesting, the experimental system is well designed and it provides enough data. However, in my opinion the authors should consider rewriting parts of the text and redesigning Figs 1,2,4,6.
Author Response
Response to Reviewer 2 Comments
Dear Editor and Reviewer:
We deeply appreciate the time and effort you have spent in reviewing our manuscript entitled “A Novel and Effective Cre/lox-based Genetic Tool for Repeated, Targeted and Markerless Gene Integration (ID: IJMS-1370457)”. We have carefully considered all the suggestions and comments, and made corresponding corrections via the "Track Changes" function in the revised manuscript. We do hope that the revisions will make our manuscript more acceptable for its publication. Herein, we addressed in details all the revisions according to the reviewer’ comments below:
Reviewer #2:
The manuscript describes application of Cre/lox system in Yarrowia lipolytica for repeated exogenous DNA integration by recycling of the genetic marker for transformants selection. The experiments are well designed and manuscript provides enough data.
Point 1: Although the authors claim: “108 - All of the tool elements were flexible and nonspecific; thus, the Cre/lox-based genetic tool was appropriate for different species.”, the tool was tested/applied only in Y. lipolytica. Therefore, such and similar statements should be removed from the manuscript. Also, the title of the manuscript should contain “… in Yarrowia lipolytica”. I also suggest removing “effective” from the title – it is understood that published genetic tool should be effective.
Response 1: Thanks for the reviewer’s valuable comments. We have deleted all statements which claim “the Cre/lox-based genetic tool was appropriate for different species”, and the title of the manuscript has been changed into “A Novel Cre/lox-based Genetic Tool for Repeated, Targeted and Markerless Gene Integration in Yarrowia lipolytica”.
Point 2: The Section “2.1. Genetic tool design and integration process analysis” is not very informative and it does not provide information on “integration process analysis” (so this should be removed from the section title). In addition, this section (2.1) is kind of a summary of other sections, especially 2.2. and 2.3.. Therefore, the text and figures in sections 2.2 and 2.3 are partially redundant with those in section 2.1.. Actually, the figures 2, 4 and 6 are redundant with the right part of the Figure 1. Taken together, my recommendation is to merge experimental system (Figs 1, 2, 4, 6) in a single section/figure with clear and nonrepetitive explanation. (Plasmids may be presented on separate fig.)
Response 2: Thanks for the reviewer’s valuable suggestion and comments. In order to eliminate as many redundant descriptions as possible, we have modified the title of section 2.1. to “Genetic tool design”, and merged the Figures 1, 2, 4, 6 to form a new Figure 1 in the revised manuscript. Meanwhile, the redundant figures, legends and statements have been deleted. Please see Figure 1 and Lines 99-157 in the revised manuscript.
Point 3: The experimental system (Figs 1, 2, 4, 6) is interesting and well designed but it can be more precisely and more clearly presented. The figures have several issues, e.g.:
Fig 1
Different terms are used in picture and in corresponding text. It is not clear what is “initial strain” (not indicated in Fig). Same for “transitional strain”. The same and convenient strain names should be used in text and Figs.
Response 3: Thanks for the reviewer’s comments. The same and convenient strain names have be used in revised manuscript. The “initial strain” has been changed into “the FY1 strain”, and the “transitional strain” has been changed into “the FY1/Cre-Y2 strain”. Please see Lines 102, 158, 212, 398 and 510 in the revised manuscript.
Point 4: 99 – “Cre-Y1 consisted of a yeast selection marker, cre gene expression cassette, bacterial element (antibiotic resistance gene and origin of replication), homologous fragment of target locus in the genome (HF-A), and two convergently oriented lox mutants. The cre expression cassette should adopt an inducible promoter to control the production of Cre recombinase. Cre-Y2 was derived from Cre-Y1 by replacing HF-A and two convergently oriented lox mutants with another homologous fragment of target locus (HF-B), a lox mutant and its reverse complement (rclox), and a target gene expression cassette. To avoid excising original segments from the genome, HF-B should be close to HF-A. Cre-Y3 was similar to Cre-Y2, but with the corresponding lox mutants of Cre-Y2 (e.g., LE for Cre-Y2 and RE for Cre-Y3).”
It should be clear that yeast selection marker is “Ura3” (are dominant and recessive genes correctly indicated? E.g. dominant URA3; recessive ura3?). Are “bacterial elements” correctly written? HF-A and HF-B are not indicated in fig etc – the text is confusing and it barely follows Fig1. I would rather use “mutated lox sequence” than “lox mutant”.
Response 4: Thanks for the reviewer’s comments. We are sorry for the incomplete characterization, and we have re-described it. Meanwhile, the “Ura3”, “bacterial elements”, “HF-A”, “HF-B” and “lox mutant” have been respectively changed into “URA3”, “Amp-ori” (antibiotic resistance gene and origin of replication), “Upleu”, “leu2” and “mutated lox sequence”. Please see Lines 99-157 in the revised manuscript.
Point 5: 118-140 – text is redundant with 99-109 as well as with Figure 1 caption.
Response 5: Thanks for the reviewer’s comment. To avoid redundancy, Lines 118-140 and Lines 99-109 in the original valuable suggestions have been deleted, and we have rewritten the section “2.1. Genetic tool design”. Please see the section 2.1. in the revised manuscript.
Point 6: Fig 2, Fig 4, Fig 6
The recombining sequences should be drawn in the same orientation (otherwise the picture is confusing)
The orientation of the plasmid sequence, after integration should be checked
The legend is mostly dispensable/redundant – sequences are indicated in upper pictures – Basically the same legends in those 3 Figs.
Response 6: Thanks for the reviewer’s comments. The recombining sequences have been drawn in the same orientation, the orientation of the plasmid sequences have been checked, and the legends have been merged. Please see the new Figure 1 in the revised manuscript.
Point 7: 49 – “gene integration” – plasmid integration or gene replacement?
Response 7: Thanks for the reviewer’s comment. The “gene integration” means “plasmid integration”.
Point 8: 52 – “gene integration efficiency “ – plasmid integration or gene replacement (ends-in or ends-out HR)? What is efficiency of targeted plasmid integration in this manuscript?
Response 8: Thanks for the reviewer’s comments. The “gene integration” means “plasmid integration”. In fact, the plasmid was first integration into the genome by homologous recombination, and the fragment containing the selection marker gene was then removed via Cre/lox-site-specific recombination. Herein, the homologous recombination-mediated gene integration efficiency is more than 80% when the length of homologous fragment is over 1 kb in Y. lipolytica (Barth, G.; Gaillardin, C. Yarrowia lipolytica. In: Nonconventional yeasts in biotechnology. Berlin, Heidelberg: Springer. 1996, 313~388)
Point 9: 54 - Besides, gene insertion via HR-mediated double-strand break (DSB) repair usually demands synchronous introduction of two plasmids (why? This should be further elaborated) harbouring different selection markers [21,22], resulting in reduction of transformation efficiency and additional procedure for plasmids recovery. – “gene insertion via HR…” – ends-in or ends-out mechanism?
Response 9: Thanks for the reviewer’s comments. We are sorry for the inaccurate description. CRISPR-Cas9 introduces a double-strand break (DSB) in the target genomic DNA, then the DSB will be repaired via non-homologous end joining (NHEJ) or HR-mediated repair with proper template. It is the reason why gene insertion via CRISPR-Cas9-based tools usually demands two plasmids, one of the plasmids containing CRISPR-Cas9-related elements, another plasmid (or fragment) provide the repair template and the target gene cassette, in the case of DSB repaired by the repair template, which results in gene insertion. This gene insertion process does not result in gene replacement, namely ends-out, and should belong to ends-in.
In order to describe it more clearly and to avoid misunderstanding, we have re-described it to “Besides, gene insertion via CRISPR-Cas9-mediated HR repair usually demands synchronous introduction of two plasmids harboring CRISPR-Cas9-related elements and repair template respectively [21,22], resulting in reduction of transformation efficiency and additional procedure to remove CRISPR-Cas9-related elements”. Please see lines 55-59 in the revised manuscript.
Point 10: 63 – “disturb further recombination” – please explain.
Response 10: Thanks for the reviewer’s comment. As stated in the manuscript, Cre recombinase can recognize and bind to loxP site, and can catalyze molecular recombination reactions including insertion, translocation, inversion and deletion. The recombination between two wild-type loxP sites will remain a new wild-type loxP site in the host, and this redundant loxP site can also be recognized by Cre recombinase and participating in recombination reaction with other loxP site, which will lead to unexpected gene rearrangement or gene deletion. That is, disturb further recombination.
Point 11: 63 – “mutational lox sites” – or mutated?
Response 11: Thanks for the reviewer’s suggestion. We have made corresponding revision. Please see Line 65 in the revised manuscript.
Point 12: 69 – “lox mutants” – or mutated lox sequences? Etc…
Response 12: Thanks for the reviewer’s suggestion. All “lox mutant” have been changed into “mutated lox sequence”. Please see Lines 71 and 273 in the revised manuscript.
Point 13: 83 – “integration, and all unnecessary fragments could be totally excised.” – does the lox site retain in the genome, is it necessary?
Response 13: Thanks for the reviewer’s comment. The lox71 and rclox66 sites retain in the genome and are necessary for next round of gene integration, after this, the generated lox72 site is dispensable for subsequent gene integration.
Point 14: 85 – “and three new plasmids were constructed.” Is this an important information in this context?
Response 14: Thanks for the reviewer’s comment. Our genetic tool comprised three plasmids, thus, we think this is an important information.
Point 15: 84 – “important lipase gene” – which gene?
Response 15: Thanks for the reviewer’s comment. “important lipase gene” has been changed into “the Rhizomucor miehei lipase gene (rml)”. Please see Lines 87-88 in the revised manuscript.
Point 16: 85 – qualitative or quantitative reporter gene?
Response 16: Thanks for the reviewer’s comment. The Rhizomucor miehei lipase activity of different recombinant strains could be determined using previously described method (Zhou, Q.; Jiao, L.; Qiao, Y.; et al. Overexpression of GRAS Rhizomucor miehei lipase in Yarrowia lipolytica via optimizing promoter, gene dosage and fermentation parameters. J. Biotechnol. 2019, 306, 16-23). And we compared the lipase activities of different strains in section 2.7., so we think rml gene is a quantitative reporter gene.
Point 17: 106 – it is not clear what are “original segments from the genome”.
Response 17: Thanks for the reviewer’s comment. “original segments from the genome” has been deleted when we rewrote the section “2.1. Genetic tool design” in the revised manuscript.
Point 18: 108 – flexible and nonspecific ?
Response 18: Thanks for the reviewer’s comment. “flexible and nonspecific” has been deleted when we rewrote the section “2.1. Genetic tool design” in the revised manuscript.
Point 19: In my opinion it is appropriate to write e.g “strain Po1f” and “plasmid Cre-Y1”, not only “Po1f” or “Cre-Y1”.
Response 19: Thanks for the reviewer’s suggestion. We have made corresponding revision. Please see Lines 169, 195, 199, 212, 270, 271, 282, 283, 378, 397, 480 and 493 in the revised manuscript.
Point 20: “target gene” and “target gene expression cassette” – in terms of genetic (homologous) recombination “target gene” is usually used for gene (or sequence) in the host genome which is targeted by integration of transforming DNA (plasmid or linear DNA fragment).
Response 20: Thanks for the reviewer’s comments. The “target gene” and “target gene expression cassette” have been changed into “rml gene”, “gene”, “rml gene expression cassette” or “gene expression cassette” depending on the context. Please see Lines 78, 94, 103, 239, 344, 386, 392, 433, 451 and 496 in the revised manuscript.
Point 21: 164 – “Ura3” – URA3 (if it is dominant, functional gene).
Response 21: Thanks for the reviewer’s suggestion. All “Ura3” has been changed into “URA3”. Please see Lines 104, 174, 179 and 408 in the revised manuscript.
Point 22: 178 –“ (about 900 bp)” – there is no such band in Fig 3.
Response 22: Thanks for the reviewer’s comment. We are sorry for this mistake, “ (about 900 bp)” has been changed into “ (about 1000 bp)”, and the molecular weight of DNA marker has been re-described. Please see Line 194 and Figure 2B in the revised manuscript.
Point 23: As I have already mentioned, the manuscript is interesting, the experimental system is well designed and it provides enough data. However, in my opinion the authors should consider rewriting parts of the text and redesigning Figs 1,2,4,6.
Response 23: Thanks for the reviewer’s comments. we rewrote the section “2.1. Genetic tool design” and merged the Figures 1, 2, 4, 6 to form a new Figure 1 in the revised manuscript. Please see Figure 1 and Lines 99-157 in the revised manuscript.
In the end, we are very grateful for the reviewer and the editor for their kindness!

Round 2
Reviewer 2 Report
The authors have accepted most of the reviewer’s suggestions but a few more corrections need to be made.
Response 4: Thanks for the reviewer’s comments. We are sorry for the incomplete characterization, and we have re-described it. Meanwhile, the “Ura3”, “bacterial elements”, “HF-A”, “HF-B” and “lox mutant” have been respectively changed into “URA3”, “Amp-ori” (antibiotic resistance gene and origin of replication), “Upleu”, “leu2” and “mutated lox sequence”. Please see Lines 99-157 in the revised manuscript.
Comments to response 4: Fig 1 – plasmids contain recessive ura3 and leu2 genes? Appropriate/correct/conventional gene/sequence nomenclature should be used in all figures, tables and text for all genes/sequences. Please use italic according to conventional nomenclature. The phenotype of all yeast strains should be indicated (e.g.: Ura+ Ura- Leu+ Leu- …..). Fig 1c – One could conclude that Po1f strain contains two leu2 alleles (Upleu and leu2-270). Is this correct?
106-107 – it should be noted that antibiotic resistance gene and origin of replication (Amp-ori) are bacterial (E. coli) DNA sequences. Please use the link below for proper gene/sequence nomenclature in figs, text and tables.
https://openwetware.org/wiki/E._coli_genotypes
107-108 – “the upstream fragment of leu2 gene (GenBank: AF260230.1, named 107 Upleu)” Is this 3’- or 5’ fragment of LEU2 gene? Length in bp? It may be denoted e.g. leu2Δ3’ or leu2Δ5’
109 – “leu2 gene” – dominant or recessive, LEU2 or leu2? What is leu2-270?
Response 7: Thanks for the reviewer’s comment. The “gene integration” means “plasmid integration”.
Comments to response 7: “Gene integration” and “plasmid integration” are not synonyms. A gene can be integrated in the genome by nonhomologous and homologous recombination which can be initiated by DSB at transforming DNA (targeted plasmid integration and gene replacement) or DSB in the genome (eg CRISPR/Cas).
Response 8: Thanks for the reviewer’s comments. The “gene integration” means “plasmid integration”. In fact, the plasmid was first integration into the genome by homologous recombination, and the fragment containing the selection marker gene was then removed via Cre/lox-site-specific recombination. Herein, the homologous recombination-mediated gene integration efficiency is more than 80% when the length of homologous fragment is over 1 kb in Y. lipolytica (Barth, G.; Gaillardin, C. Yarrowia lipolytica. In: Nonconventional yeasts in biotechnology. Berlin, Heidelberg: Springer. 1996, 313~388)
Comments to response 8: Please see Comments to response 7. Appropriate text/explanation should be added to the manuscript.
Response 9: Thanks for the reviewer’s comments. We are sorry for the inaccurate description. CRISPR-Cas9 introduces a double-strand break (DSB) in the target genomic DNA, then the DSB will be repaired via non-homologous end joining (NHEJ) or HR-mediated repair with proper template. It is the reason why gene insertion via CRISPR-Cas9-based tools usually demands two plasmids, one of the plasmids containing CRISPR-Cas9-related elements, another plasmid (or fragment) provide the repair template and the target gene cassette, in the case of DSB repaired by the repair template, which results in gene insertion. This gene insertion process does not result in gene replacement, namely ends-out, and should belong to ends-in.
In order to describe it more clearly and to avoid misunderstanding, we have re-described it to “Besides, gene insertion via CRISPR-Cas9-mediated HR repair usually demands synchronous introduction of two plasmids harboring CRISPR-Cas9-related elements and repair template respectively [21,22], resulting in reduction of transformation efficiency and additional procedure to remove CRISPR-Cas9-related elements”. Please see lines 55-59 in the revised manuscript.
Comments to response 9: 55 – 59 – “CRISPR-Cas9-mediated HR repair” probably should be rewrite in “homologous recombination initiated by repair od DSB introduced by CRISPR-Cas9. The template for homologous recombination initiated by CRISPR-Cas9 is usually linear DNA fragment such as PCR product.
Response 10: Thanks for the reviewer’s comment. As stated in the manuscript, Cre recombinase can recognize and bind to loxP site, and can catalyze molecular recombination reactions including insertion, translocation, inversion and deletion. The recombination between two wild-type loxP sites will remain a new wild-type loxP site in the host, and this redundant loxP site can also be recognized by Cre recombinase and participating in recombination reaction with other loxP site, which will lead to unexpected gene rearrangement or gene deletion. That is, disturb further recombination.
Comments to response 10: “disturb further recombination” is not enough to understand the point. It is obvious that presence of lox sites in the genome can initiate inter- and intrachromosomal recombination resulting in genome instability, and also make strain construction more difficult. Etc
“molecular recombination reactions” – “recombination” or “genetic recombination” is enough; resulting in inversion……
Response 12: Thanks for the reviewer’s suggestion. All “lox mutant” have been changed into “mutated lox sequence”. Please see Lines 71 and 273 in the revised manuscript.
Comments to response 12: 304, 307-308 – “LE/RE 307 mutants” – it seems that there is no explanation for LE/RE (mutants?)
Response 13: Thanks for the reviewer’s comment. The lox71 and rclox66 sites retain in the genome and are necessary for next round of gene integration, after this, the generated lox72 site is dispensable for subsequent gene integration. (Point 13: 83 – “integration, and all unnecessary fragments could be totally excised.” – does the lox site retain in the genome, is it necessary?)
Comments to response 13: The reviewer’s point might be that in final strain at least one lox site retains in the genome so it is not totally correct that “all unnecessary fragments could be totally excised”.
Response 16: Thanks for the reviewer’s comment. The Rhizomucor miehei lipase activity of different recombinant strains could be determined using previously described method (Zhou, Q.; Jiao, L.; Qiao, Y.; et al. Overexpression of GRAS Rhizomucor miehei lipase in Yarrowia lipolytica via optimizing promoter, gene dosage and fermentation parameters. J. Biotechnol. 2019, 306, 16-23). And we compared the lipase activities of different strains in section 2.7., so we think rml gene is a quantitative reporter gene.
Comments to response 16: 88 – “qualitative reporter” – the authors emphasized that the gene is qualitative reporter marker. So, it seems that the reviewer, therefore, ask whether the lipase gene is qualitative or quantitative reporter gene. Please delete qualitative (almost all genetic markers are qualitative, so there is no need to stress that), or If it is qualitative genetic marker please explain in the manuscript why is it so, and why is that important.
Response 22: Thanks for the reviewer’s comment. We are sorry for this mistake, “ (about 900 bp)” has been changed into “ (about 1000 bp)”, and the molecular weight of DNA marker has been re-described. Please see Line 194 and Figure 2B in the revised manuscript.
Comments to response 22: 194 – please check the size of all bands in all figs
Additional comments
36-37 – it is not clear what is tandem fragment; is it possible to inset one copy of transforming DNA in the genome of Y.lipolytica? Although the references are provided, I would appreciate a short explanation of zeta loci.
77 – reference 34 – the term “tedious procedure” might not be appropriate; I am not convinced that the procedure described in this manuscript “comprising three plasmids” is less tedious. Is there anything worthwhile in reference 34?
93 – “six different genes” - which genes?
94 – “another target gene” which gene?
96 – I would not use “gene editing” – rather “targeted modification of the genome” – please check throughout of the manuscript
167 – “auxotrophic host” phenotype, Ura-, Leu-?
168 – “in 5 mL of MD-Leu medium to enrich the Ura+ recombinant” – please check is this correct. Please check MO and MD media throughout manuscript. MO-Leu-Ura, MO-Ura, MO. What is MO, is it the same as MO-Leu-Ura?
304 – “combination” or “recombination” Please, check English expression throughout all manuscript.
304-305 – “increasingly less recombination reactions” – maybe you should use recombination frequency?
383 – “varied resources” – ? or organisms, or origins? Actually, only from (homologous?) Y.lipolytica and Vitreoscilla
386 – integration times - ?
399 and 497 – Upaxp (locus) – no explanation in the manuscript. Genetic nomenclature?
Author Response
Response to Reviewer 2 Comments
Dear Editor and Reviewer:
We deeply appreciate the time and effort you have spent in reviewing our manuscript entitled “A Novel Cre/lox-based Genetic Tool for Repeated, Targeted and Markerless Gene Integration in Yarrowia lipolytica (ID: IJMS-1370457)”. We have carefully considered all the suggestions and comments, and made corresponding corrections via the "Track Changes" function in the revised manuscript. We do hope that the revisions will make our manuscript more acceptable for its publication. Herein, we addressed in details all the revisions according to the reviewer’ comments below:
Reviewer #2:
Point 1: Response 4: Thanks for the reviewer’s comments. We are sorry for the incomplete characterization, and we have redescribed it. Meanwhile, the “Ura3”, “bacterial elements”, “HF-A”, “HF-B” and “lox mutant” have been respectively changed into “URA3”, “Amp-ori” (antibiotic resistance gene and origin of replication), “Upleu”, “leu2” and “mutated lox sequence”. Please see Lines 99-157 in the revised manuscript.
Comments to response 4: Fig 1 – plasmids contain recessive ura3 and leu2 genes? Appropriate/correct/conventional gene/sequence nomenclature should be used in all figures, tables and text for all genes/sequences. Please use italic according to conventional nomenclature. The phenotype of all yeast strains should be indicated (e.g.: Ura+ Ura- Leu+ Leu- …..). Fig 1c – One could conclude that Po1f strain contains two leu2 alleles (Upleu and leu2-270). Is this correct?
Response 1: Thanks for the reviewer’s valuable suggestion and comments. The plasmids contain dominant ura3 and leu2 genes, and their corresponding recessive genes are ura3-302 and leu2-270 in Y. lipolytica Po1f, respectively. To avoid being misunderstood, we have changed the dominant gene (ura3 and leu2) to uppercase (URA3 and LEU2), and used italic according to conventional nomenclature. In addition, the phenotype of all yeast strains is indicated when these strains first appeared in the article.
In fact, the Po1f strain contains only one LEU2 alleles gene, which is defective leu2-270. As described in the manuscript, Upleu is the upstream fragment of leu2-270 gene rather than the allele of the LEU2 gene, and is used as a homologous fragment when the vector (Cre-Y1) is integrated into the host.
Point 2: 106-107 – it should be noted that antibiotic resistance gene and origin of replication (Amp-ori) are bacterial (E. coli) DNA sequences. Please use the link below for proper gene/sequence nomenclature in figs, text and tables.
https://openwetware.org/wiki/E._coli_genotypes
Response 2: Thanks for the reviewer’s valuable suggestion and comments. In fact, the Amp-ori fragment contains ampicillin resistance gene and origin of replication in E. coli. For ease of description, we named it Amp-ori. In order to be more standardized, we redescribed the Amp-ori fragment in the revised manuscript. Please see Lines 114-115 in the revised manuscript.
Point 3: 107-108 – “the upstream fragment of leu2 gene (GenBank: AF260230.1, named 107 Upleu)” Is this 3’- or 5’ fragment of LEU2 gene? Length in bp? It may be denoted e.g. leu2Δ3’ or leu2Δ5’.
Response 3: Thanks for the reviewer’s comments. Upleu fragment is located at the upstream region of leu2-270 gene, it is not part of LEU2 gene. Its length is 964 bp, and we have added the length. Please see Line 115 in the revised manuscript.
Based on the characteristics of the fragment, we think Upleu is more appropriate than leu2Δ5. “Δ” is usually used for recessive gene, but Upleu is just a DNA sequence.
Point 4: 109 – “leu2 gene” – dominant or recessive, LEU2 or leu2? What is leu2-270?
Response 4: Thanks for the reviewer’s comments. “leu2 gene” is dominant, leu2-270 is recessive leu2 allele. To avoid being misunderstood, we have changed the dominant gene (ura3 and leu2) to uppercase (URA3 and LEU2), and used italic according to conventional nomenclature. Please see Line 116 in the revised manuscript.
Point 5: Response 7: Thanks for the reviewer’s comment. The “gene integration” means “plasmid integration”.
Comments to response 7: “Gene integration” and “plasmid integration” are not synonyms. A gene can be integrated in the genome by nonhomologous and homologous recombination which can be initiated by DSB at transforming DNA (targeted plasmid integration and gene replacement) or DSB in the genome (eg CRISPR/Cas).
Response 8: Thanks for the reviewer’s comments. The “gene integration” means “plasmid integration”. In fact, the plasmid was first integrated into the genome by homologous recombination, and the fragment containing the selection marker gene was then removed via Cre/lox-site-specific recombination. Herein, the homologous recombination-mediated gene integration efficiency is more than 80% when the length of homologous fragment is over 1 kb in Y. lipolytica (Barth, G.; Gaillardin, C. Yarrowia lipolytica. In: Nonconventional yeasts in biotechnology. Berlin, Heidelberg: Springer. 1996, 313~388)
Comments to response 8: Please see Comments to response 7. Appropriate text/explanation should be added to the manuscript.
Response 5: Thanks for the reviewer’s valuable suggestion and comments. We are sorry for the inaccurate description. In deed, “Gene integration” and “plasmid integration” are caused by different repair mechanisms. Following your suggestion, we have added specific details and redescribed it to “Another study accomplished gene integration into PEX10 locus in Y. lipolytica via a dual CRISPR-Cas9 cleavage strategy directed by paired sgRNAs, the integration efficiencies of homologous recombination (HR) based method uses a circular plasmid as the repair template and homology-mediated end-joining based method using a linear template were 16.7%±3.6% and 37.5%±8.8%, respectively”. Please see Lines 52-57 in the revised manuscript.
Point 6: Response 9: Thanks for the reviewer’s comments. We are sorry for the inaccurate description. CRISPR-Cas9 introduces a double-strand break (DSB) in the target genomic DNA, then the DSB will be repaired via non-homologous end joining (NHEJ) or HR-mediated repair with proper template. It is the reason why gene insertion via CRISPR-Cas9-based tools usually demands two plasmids, one of the plasmids containing CRISPR-Cas9-related elements, another plasmid (or fragment) provide the repair template and the target gene cassette, in the case of DSB repaired by the repair template, which results in gene insertion. This gene insertion process does not result in gene replacement, namely ends-out, and should belong to ends-in.
In order to describe it more clearly and to avoid misunderstanding, we have redescribed it to “Besides, gene insertion via CRISPR-Cas9-mediated HR repair usually demands synchronous introduction of two plasmids harboring CRISPR-Cas9-related elements and repair template respectively [21,22], resulting in reduction of transformation efficiency and additional procedure to remove CRISPR-Cas9-related elements”. Please see Lines 55-59 in the revised manuscript.
Comments to response 9: 55 – 59 – “CRISPR-Cas9-mediated HR repair” probably should be rewrite in “homologous recombination initiated by repair or DSB introduced by CRISPR-Cas9”. The template for homologous recombination initiated by CRISPR-Cas9 is usually Linear DNA fragment such as PCR product.
Response 6: Thanks for the reviewer’s suggestion. We have made corresponding revision. Please see Lines 60-62 in the revised manuscript.
Point 7: Response 10: Thanks for the reviewer’s comment. As stated in the manuscript, Cre recombinase can recognize and bind to loxP site, and can catalyze molecular recombination reactions including insertion, translocation, inversion and deletion. The recombination between two wild-type loxP sites will remain a new wild-type loxP site in the host, and this redundant loxP site can also be recognized by Cre recombinase and participating in recombination reaction with other loxP site, which will lead to unexpected gene rearrangement or gene deletion. That is, disturb further recombination.
Comments to response 10: “disturb further recombination” is not enough to understand the point. It is obvious that presence of lox sites in the genome can initiate inter- and intrachromosomal recombination resulting in genome instability, and make strain construction more difficult. Etc
Response 7: Thanks for the reviewer’s comments. “disturb further recombination” has been changed into “resulting in genome instability, and also making strain construction more difficult”. Please see Lines 71-72 in the revised manuscript.
Besides, in our multi-round integration processes, all lox sites are on the same chromosome (nearby LEU2 locus), and they cannot initiate inter- and intrachromosomal recombination, which need lox sites are on different chromosomes.
Point 8: “molecular recombination reactions” – “recombination” or “genetic recombination” is enough; resulting in inversion……
Response 8: Thanks for the reviewer’s comment. “molecular recombination reactions” has been changed into “recombination”. Please see Line 69 in the revised manuscript.
Point 9: Response 12: Thanks for the reviewer’s suggestion. All “lox mutant” have been changed into “mutated lox sequence”. Please see Lines 71 and 273 in the revised manuscript.
Comments to response 12: 304, 307-308 – “LE/RE mutants” – it seems that there is no explanation for LE/RE (mutants?)
Response 9: Thanks for the reviewer’s comments. We are sorry for the inaccurate description here. Actually, “LE/RE mutants” means “LE or RE mutants”. Hence, “LE/RE mutants” has been changed into “LE or RE sites”. Please see Lines 315 and 318 in the revised manuscript.
Point 10: Response 13: Thanks for the reviewer’s comment. The lox71 and rclox66 sites retain in the genome and are necessary for next round of gene integration, after this, the generated lox72 site is dispensable for subsequent gene integration. (Point 13: 83 – “integration, and all unnecessary fragments could be totally excised.” – does the lox site retain in the genome, is it necessary?)
Comments to response 13: The reviewer’s point might be that in final strain at least one lox site retains in the genome so it is not totally correct that “all unnecessary fragments could be totally excised”.
Response 10: Thanks for the reviewer’s suggestion. we have rewritten the statements which claim “all unnecessary fragments could be totally excised”. Please see Lines 18, 93, 227, 292 and 465 in the revised manuscript.
Point 11: Response 16: Thanks for the reviewer’s comment. The Rhizomucor miehei lipase activity of different recombinant strains could be determined using previously described method (Zhou, Q.; Jiao, L.; Qiao, Y.; et al. Overexpression of GRAS Rhizomucor miehei lipase in Yarrowia lipolytica via optimizing promoter, gene dosage and fermentation parameters. J. Biotechnol. 2019, 306, 16-23). And we compared the lipase activities of different strains in section 2.7., so we think rml gene is a quantitative reporter gene.
Comments to response 16: 88 – “qualitative reporter” – the authors emphasized that the gene is qualitative reporter marker. So, it seems that the reviewer, therefore, ask whether the lipase gene is qualitative or quantitative reporter gene. Please delete qualitative (almost all genetic markers are qualitative, so there is no need to stress that), or If it is qualitative genetic marker please explain in the manuscript why is it so, and why is that important.
Response 11: Thanks for the reviewer’s suggestion. We have deleted “qualitative”. Please see Line 95 in the revised manuscript.
Point 12: Response 22: Thanks for the reviewer’s comment. We are sorry for this mistake, “ (about 900 bp)” has been changed into “ (about 1000 bp)”, and the molecular weight of DNA marker has been redescribed. Please see Line 194 and Figure 2B in the revised manuscript.
Comments to response 22: 194 – please check the size of all bands in all figs
Response 12: Thanks for the reviewer’s comment. We have checked the size of all bands in all figures.
Point 13: 36-37 – it is not clear what is tandem fragment; is it possible to inset one copy of transforming DNA in the genome of Y.lipolytica? Although the references are provided, I would appreciate a short explanation of zeta loci.
Response 13: Thanks for the reviewer’s comments. Actually, “tandem fragment of target gene and selection marker” in the original manuscript means “fusion fragment of target gene and selection marker”. To avoid being misunderstood, we have changed it into “gene fragment and selection marker”. Please see Lines 36-37 in the revised manuscript.
It is possible to insert one copy or multi-copy of transforming DNA in the genome of Y.lipolytica, which greatly depends on the screening marker and screening pressure.
In addition, “zeta” is the long terminal repeated sequence on the retrotransposon Ylt1 of Y. lipolytica, its length is 714 bp. There are at least 35 copies of Ylt1 retrotransposon present in each genome in a dispersed manner in Ylt1-carrying hosts. The zeta sequences thus provide many targeting sites, which provides a useful tool for integration of multiple-copy vectors. Follow your suggestion, we have added more details about “zeta” loci. Please see Lines 37-39 in the revised manuscript.
Point 15: 77 – reference 34 – the term “tedious procedure” might not be appropriate; I am not convinced that the procedure described in this manuscript “comprising three plasmids” is less tedious. Is there anything worthwhile in reference 34?
Response 15: Thanks for the reviewer’s comment. “tedious” has been deleted in the revised manuscript. Please see Line 89 in the revised manuscript.
Reference 34 combined Cre/lox system and 26S rDNA to develop a versatile framework to iteratively integrate multicopy metabolic pathways in Y. lipolytica, and the iterative integration is happening at 80% rates. We think it is an inspired method.
Point 16: 93 – “six different genes” - which genes?
Response 16: Thanks for the reviewer’s comment. “six different genes” are the ire1, kar2, pdi, sls1, hac1, and vgb genes. Please see Section 2.9 or Lines 509-510 in the manuscript. For ease of understanding, we have added the details of six different genes. Please see Line 100 in the revised manuscript.
Point 17: 94 – “another target gene” which gene?
Response 17: Thanks for the reviewer’s comment. “another target gene” have been changed into “another gene (Rhizopus oryzae lipase gene, rol)”. Please see Line 102 in the revised manuscript.
Point 18: 96 – I would not use “gene editing” – rather “targeted modification of the genome” – please check throughout of the manuscript
Response 18: Thanks for the reviewer’s suggestion. We have made corresponding revision. Please see Line 104 in the revised manuscript.
Point 19: 167 – “auxotrophic host” phenotype, Ura-, Leu-?
Response 19: Thanks for the reviewer’s comment. Auxotrophic host Po1f is Leu- and Ura-. Please see Table S2 in the supplementary file. And we have added the phenotype of all yeast strains.
Point 20: 168 – “in 5 mL of MD-Leu medium to enrich the Ura+ recombinant” – please check is this correct. Please check MO and MD media throughout manuscript. MO-Leu-Ura, MO-Ura, MO. What is MO, is it the same as MO-Leu-Ura?
Response 20: Thanks for the reviewer’s suggestion. Since Ura+ recombinant can grow in the medium without uracil, so “in 5 mL of MD-Leu medium to enrich the Ura+ recombinant” is correct.
We have checked MO and MD media in the revised manuscript.
The MO medium (20 mL/L oleic acid, 0.5 mL/L Tween 80, 13.4 g/L yeast nitrogen base with no amino acids, and 0.4 mg/L biotin) was derived from MD medium, and leucine (final concentration, 262 mg/L) or uracil (final concentration, 22.4 mg/L) was added for auxotrophic strains. Please see Lines 483-486 in the revised manuscript.
MO-Leu-Ura medium: MO medium supplemented with leucine and uracil. MO-Ura medium: MO medium supplemented with uracil. We have added corresponding description. Please see Lines 180 and 255 in the revised manuscript.
Point 21: 304 – “combination” or “recombination” Please, check English expression throughout all manuscript.
Response 21: Thanks for the reviewer’s comment. The Cre recombinase can recognize and combinate loxP site, and then catalyze recombination. In order to avoid unreadable description, “combination” has been changed into “recognition”. Please see Line 315 in the revised manuscript.
Point 22: 304-305 – “increasingly less recombination reactions” – maybe you should use recombination frequency?
Response 22: Thanks for the reviewer’s comments. We have made corresponding revision. Please see Line 316 in the revised manuscript.
Point 23: 383 – “varied resources” – ? or organisms, or origins? Actually, only from (homologous?) Y.lipolytica and Vitreoscilla.
Response 23: Thanks for the reviewer’s comment. “varied lengths and resources” have been changed into “varied lengths and two resources”. Please see Line 394 in the revised manuscript.
Point 24: 386 – integration times - ?
Response 24: Thanks for the reviewer’s comment. We are sorry for the incomplete characterization. To avoid misunderstanding, we have redescribed it. Please see Line 397-398 in the revised manuscript.
Point 25: 399 and 497 – Upaxp (locus) – no explanation in the manuscript. Genetic nomenclature?
Response 25: Thanks for the reviewer’s comments. Upaxp is located at the upstream region of AXP gene. We have added the details to describe the Upaxp. Please see Lines 407-408 in the revised manuscript.
In the end, we are very grateful for the reviewer and the editor for their kindness!
